# Red Blood Cell BCL-x_L_ Is Required for *Plasmodium falciparum* Survival: Insights into Host-Directed Malaria Therapies

**DOI:** 10.3390/microorganisms10040824

**Published:** 2022-04-15

**Authors:** Coralie Boulet, Ghizal Siddiqui, Taylah L. Gaynor, Christian Doerig, Darren J. Creek, Teresa G. Carvalho

**Affiliations:** 1Department of Microbiology, Anatomy, Physiology and Pharmacology, School of Agriculture, Biomedicine and Environment, La Trobe University, Bundoora, VIC 3086, Australia; coralie.boulet@burnet.edu.au (C.B.); 18498207@students.latrobe.edu.au (T.L.G.); 2Monash Institute of Pharmaceutical Sciences, Monash University, Parkville, VIC 3052, Australia; ghizal.siddiqui@monash.edu (G.S.); darren.creek@monash.edu (D.J.C.); 3School of Health and Biomedical Sciences, RMIT University, Bundoora, VIC 3083, Australia; christian.doerig@rmit.edu.au

**Keywords:** BCL-x_L_, malaria, *Plasmodium falciparum*, host-directed therapy, red blood cells, host–parasite interaction

## Abstract

The development of antimalarial drug resistance is an ongoing problem threatening progress towards the elimination of malaria, and antimalarial treatments are urgently needed for drug-resistant malaria infections. Host-directed therapies (HDT) represent an attractive strategy for the development of new antimalarials with untapped targets and low propensity for resistance. In addition, drug repurposing in the context of HDT can lead to a substantial decrease in the time and resources required to develop novel antimalarials. Host BCL-x_L_ is a target in anti-cancer therapy and is essential for the development of numerous intracellular pathogens. We hypothesised that red blood cell (RBC) BCL-x_L_ is essential for *Plasmodium* development and tested this hypothesis using six BCL-x_L_ inhibitors, including one FDA-approved compound. All BCL-x_L_ inhibitors tested impaired proliferation of *Plasmodium falciparum* 3D7 parasites *in vitro* at low micromolar or sub-micromolar concentrations. Western blot analysis of infected cell fractions and immunofluorescence microscopy assays revealed that host BCL-x_L_ is relocated from the RBC cytoplasm to the vicinity of the parasite upon infection. Further, immunoprecipitation of BCL-x_L_ coupled with mass spectrometry analysis identified that BCL-x_L_ forms unique molecular complexes with human μ-calpain in uninfected RBCs, and with human SHOC2 in infected RBCs. These results provide interesting perspectives for the development of host-directed antimalarial therapies and drug repurposing efforts.

## 1. Introduction

Malaria is a vector-borne parasitic infection caused by apicomplexan parasites of the genus *Plasmodium*, of which *Plasmodium falciparum* causes the most severe forms of disease. Malaria remains an ongoing and significant global health threat in the 21st century. Recently, disruptions caused by the COVID-19 pandemic further increased the impact of malaria, with 241 million cases and 627,000 deaths in 2020 (a 12% increase compared to 2019 [1]). Despite significant efforts made towards malaria control in recent decades, many formidable challenges stand in the way of the elimination of malaria [2]. This includes the emergence and spread of drug resistance, the low efficacy of the recently approved malaria vaccine [3,4], the impact of the climate crisis on health care systems and vector distribution, and additional pandemics. Worryingly, reduced susceptibility of parasites to the current frontline treatment, artemisinin-combination therapies (ACTs), is also observed [5]. 

*P. falciparum* initially invades and replicates inside liver cells, but disease symptoms and deaths associated with malaria are caused by the subsequent parasite replication inside red blood cells (RBCs); therefore, most current antimalarials target blood-stage parasite molecules. Noteworthy, as drug resistant parasites develop against all existing antimalarial treatments, host-directed therapies (HDT) emerge as a novel and promising antimalarial strategy for two main reasons [6,7]. Firstly, targeting host molecules that are required for the development of *Plasmodium* limits the most straightforward pathway for drug resistance (i.e., mutation of the target molecule). Secondly, many drugs targeting human molecules already exist in the context of other diseases such as cancers [8] and may be repurposed for antimalarial treatment, therefore saving time and resources required for drug development.

In this perspective, host programmed cell death pathways represent potential anti-parasitic targets. This is because apicomplexan parasites, including *Plasmodium* spp., are known to manipulate the host cell death pathways to allow their own intracellular survival [9,10]. Further, many compounds targeting cell death pathways are available and are clinically approved for the treatment of other human diseases [8]. Importantly, this host-directed strategy has proven effective against *Plasmodium* parasites: ABT-737, an inhibitor of the anti-apoptotic members of the BCL-2 (B-cell lymphoma 2) family BCL-2 and BCL-x_L_, was shown to eliminate liver stages of *Plasmodium* spp. both *in vitro* and in murine models [11]. Importantly, no *bcl-2* or *bcl-x_L_* orthologs have been identified in *Plasmodium* spp., suggesting that the effect observed was due to the inhibition of the host molecules (although parasite-encoded off-targets cannot be excluded at this stage). RBCs are devoid of mitochondria and are therefore unable to undergo classical apoptosis. However, BCL-x_L_ has been identified in proteomics studies of RBCs [12], and its inhibition induces eryptosis, the programmed cell death of RBCs [13,14].

The present study investigates the impact of six BCL-x_L_ inhibitors on the proliferation of *P. falciparum in vitro* and on RBCs death. The six BCL-x_L_ inhibitors include one FDA-approved drug and have been selected based on their target specificity, clinical use, and/or ongoing clinical trials. Further, the localization and binding partners of RBC BCL-x_L_ were determined in *P. falciparum*-infected RBCs (iRBCs) and uninfected RBCs (uRBCs). All BCL-x_L_ inhibitors impaired *P. falciparum* growth *in vitro* (some with submicromolar IC_50_ values) and induced eryptosis of iRBCs. Moreover, in iRBCs, BCL-x_L_ was recruited to the vicinity of the parasite and appeared to form unusual complexes, with a shift in binding partners (compared to uRBCs) upon infection. Notably, we show that BCL-x_L_ binds to μ-calpain in uRBCs and to the human SHOC2 leucine rich repeat scaffold protein in iRBCs. These observations raise fascinating questions regarding the role of BCL-x_L_ in RBCs in general, and, more crucially, during *Plasmodium* infection. Overall, these results provide exciting perspectives regarding host-directed antimalarial therapies and drug-repurposing opportunities.

## 2. Materials and Methods

### 2.1. P. falciparum Culture

*P. falciparum* 3D7 parasites were cultured *in vitro* with human RBCs (kindly donated by the Australian Red Cross) as previously described [15]. Briefly, parasites were cultured at 4% haematocrit in complete RPMI (cRPMI: 16.2 g/L RPMI 1640 Medium HEPES, 50 mg/L hypoxanthine, 10 mg/L gentamicin, 2.25 g/L sodium bicarbonate, 5 g/L AlbuMAX lipid-rich BSA), and incubated at 37 °C in 1% O_2_, 5% CO_2_, and 94% N_2_. This work was supported by an Australian Red Cross Blood Service Agreement (19-05VIC-01) and approved by the La Trobe University Research Ethics Committee (HEC17-013).

### 2.2. Growth Inhibition Assays (IC_50_)

Growth inhibition assays were conducted as previously described [16] for the following BCL-x_L_ inhibitors (Table 1): ABT-737 (Sapphire Bioscience #A10255), ABT-263 (Sapphire Bioscience #A10022), ABT-199 (Sapphire Bioscience #A12500), WEHI-539 (Sapphire Bioscience #S7100), A-1155463 (Sapphire Bioscience #S7800), and A-1331852 (Sapphire Bioscience #S7801). Asynchronous parasite stages were exposed to 0.2 nM–50 μM of compound for 72 h. Artemisinin (ART) and chloroquine (CQ) (both at 50 μM) served as a growth inhibition positive control, and DMSO as a negative control. Following a freeze–thaw cycle, parasite DNA was stained with the fluorescent nucleic acid dye SYBR Gold (Invitrogen), and fluorescence was measured on a CLARIOstar^®^ microplate reader. Data were transformed in Excel using the formula below (“averageFluo” refers to the average fluorescence over technical triplicates), and percent inhibition was graphed against the logarithm of concentration using GraphPad Prism^®^ 8 software. Data are shown as the mean +/− SD of 3 independent replicates, and the half-maximal inhibitory concentration (IC_50_) is indicated.
$$\frac{averageFluo\left(sample\right)-averageFluo\left(DMSO\right)}{averageFluo\left(ART\&CQ\right)-averageFluo\left(DMSO\right)}*100$$

### 2.3. Immunofluorescence Assay (IFA)

Cells from an asynchronous culture at high parasitemia (>5%) were fixed for 30 min in 4% paraformaldehyde and 0.0075% glutaraldehyde. Following membrane permeabilization with 0.1% Triton X-100 for 10 min at room temperature, cells were blocked with 3% Bovine Serum Albumin (BSA) overnight at 4 °C and incubated successively with primary and secondary antibodies, overnight at 4 °C and 1 h at room temperature, respectively. Cells were mixed with VectaShield containing DAPI (2 μg/mL) and imaged on a Zeiss Axioscope microscope using an Orca Digitalcamera (1000 magnification).

### 2.4. Flow Cytometry Assay

RBC death was assessed by flow cytometry, measuring the exposure of phosphatidylserine (PS) on the cell surface as previously established [16]. Briefly, asynchronous *P. falciparum* cultures and uRBCs were incubated with a BCL-x_L_ inhibitor (5 µM for compounds with a *P. falciparum* IC_50_ < 5 µM, and 10 µM for the others) for 4 h in incomplete RPMI (iRPMI; media without Albumax and sodium bicarbonate) at 37 °C under low-oxygen conditions. BCL-x_L_ inhibitors were removed by washing cells once with Phosphate-Buffered Saline (PBS), and parasite RNA and DNA were stained for 40 min with 4 µM Hoechst 33342 in Thiazole Orange (BD ReticCount™). Cells were washed in staining buffer (5 mM CaCl_2_, 5 mM glucose, 32 mM HEPES pH 7.4, 5 mM KCl, 1 mM MgSO_4_, 125 mM NaCl) and incubated for 20 min with Annexin V-PE (1:20 dilution in staining buffer; BD Bioscience) for the staining of surface-exposed PS. A total of 100,000 events were recorded on a CytoFLEX S flow cytometer (Beckman Coulter, Brea, CA, USA), and data were analysed using FlowJo V10.

### 2.5. Fractionation of Parasite and RBC Proteins

**Trophozoite-iRBC enrichment**. *P. falciparum* cultures were synchronized to ring-stage parasites by sorbitol treatment as previously described [17] and allowed to mature into trophozoites for 24 h. Magnetic separation (VarioMACS™ Separator, Miltenyi Biotec, Bergisch Gladbach, Germany) was used to purify the trophozoite-stage parasites as described previously [18]. Briefly, trophozoite cultures were loaded on an CS Column (Miltenyi Biotec, Bergisch Gladbach, Germany) previously equilibrated with iRPMI and then abundantly washed with iRPMI. The column was detached from the VarioMACS, and magnet-bound parasites were eluted into a clean tube with cRPMI. The efficiency of trophozoite enrichment was assessed by a thin blood smear.

**Total protein extraction.** Cell numbers were determined with a haemocytometer (Merck, Darmstadt, Germany). Cell lysis of 10^8^ uRBCs and 10^8^ magnet-purified iRBCs was achieved by incubation with 80 μL of M-PER™ (ThermoFisher, Waltham, MA, USA) supplemented with protease and phosphatase inhibitors (PPI) (1X Protease Inhibitors Cocktail EDTA-free (Roche, Basel, Switzerland), 20 mM sodium fluoride, 100 μM sodium orthovanadate, 1 mM PMSF, 10 mM β-glycerophosphate) on ice for 10 min, and TALON^®^ Metal Affinity Resin (Takara, Kusatsu, Japan) was used to remove the haemoglobin, as described previously [19]. Briefly, 100 μL of resin were incubated with the cell lysate for 10 min at 4 °C. Following centrifugation, the supernatant (i.e., haemoglobin-depleted protein sample) was collected, denatured with Laemmli Buffer for 5 min at 98 °C, and stored at −20 °C.

**RBC cytosolic protein vs. membrane-bound and parasite protein extraction**. Saponin is a gentle detergent that solubilizes the RBC and the parasitophorous vacuole membranes but not the parasite plasma membrane [20]. Therefore, in iRBCs, proteins localized in the RBC cytosol were separated from the fraction containing the parasite and the membrane-bound proteins through a saponin lysis. Similarly, in uRBCs, cytosolic proteins were separated from membrane proteins with the same process. Magnet-purified iRBCs and uRBCs (10^8^ cells) were incubated with 0.1% saponin-PPI on ice for 10 min, then centrifuged for 5 min at 16,200× *g* at 4 °C. The soluble fraction (i.e., supernatant) was collected, haemoglobin-depleted with TALON resin, denatured with Laemmli Buffer for 5 min at 98 °C, and stored at −20 °C (see above). The insoluble fraction (i.e., pellet of the saponin lysis, containing the parasite and membrane proteins fraction) was lysed again with 0.1% saponin-PPI on ice for 10 min, then centrifuged for 5 min at maximum speed at 4 °C. To ensure removal of haemoglobin and other RBC cytosolic proteins, the resulting pellet was washed with PBS-PPI. Samples were denatured in Laemmli Buffer for 5 min at 98 °C and stored at −20 °C.

### 2.6. Western Blot Analysis

Protein extracts of 10^7^ of either uRBCs or iRBCs and of the breast cancer cell line MDA-MB 231 (used as a positive control and kindly provided by Dr. Delphine Merino, Olivia Newton-John Cancer Research Centre) were loaded on a 4–12% Bis-Tris gel (Novex^TM^) and resolved in MES-SDS running buffer (Novex^TM^). Proteins were transferred onto a 0.2 μm nitrocellulose membrane (BioRad, Hercules, CA, USA) in transfer buffer (0.3% Tris base, 0.75% glycine, 20% methanol), and the membrane was washed in PBST (0.1% Tween-20 in PBS) for 10 min and then blocked overnight at 4 °C in 5% milk-PBST. Primary and secondary antibodies were incubated successively, overnight at 4 °C and for 1 h at room temperature, respectively. Dilutions of primary antibodies (in 5% BSA-PBST) were used as follows: BCL-x_L_ 1:1000 (Cell Signaling, Danvers, MA, USA, #2764), Carbonic Anhydrase 1:4000 (Abcam, Cambridge, UK, #ab108367), PfHSP70.1 1:5000 (kindly provided by Prof Gilson and Prof Crabb, Burnet Institute), and protein 4.1 1:500 (kindly provided by Dr. Proellocks and Prof Cooke, Monash University). Secondary antibody: anti-rabbit-HRP 1:5000 (Cell Signaling, #7074). HRP signal was revealed with Pierce™ ECL Western Blotting Substrate (Thermo Scientific) and detected with a ChemiDoc XRS+ system.

### 2.7. Immunoprecipitation of Host BCL-x_L_

**Sample preparation**. Magnet-purified trophozoite-iRBCs and uRBCs (10^9^ cells) were incubated with either 10 μM WEHI-539 or DMSO for 4 h at 37 °C in low-oxygen conditions. Cells were lysed with 800 μL M-PER™-PPI for 15 min as described above.

**Immunoprecipitation of BCL-x_L_**. Protein A magnetic beads (Gen-Script, Piscataway, NJ, USA) were pre-incubated with anti-BCL-x_L_ antibody (1:100, Cell Signaling, #2764) in TBST (20 mM Tris, 150 mM NaCl, 0.1% Tween-20) overnight at 4 °C under gentle agitation. Magnetic beads were incubated in TBST without antibody for the control (beads alone). Unbound antibody was removed with a TBST-PPI wash, and protein extracts were added to the beads and incubated overnight at 4 °C under gentle agitation. Unbound proteins were removed with TBST-PPI washing. BCL-x_L_ bound proteins were solubilized following resuspension of the washed beads in Laemmli buffer for 5 min at 95 °C. Samples were kept at −20 °C before mass spectrometry analysis. The presence of BCL-x_L_ in the immunoprecipitates was confirmed by Western Blot analysis of 1/10th of each sample.

### 2.8. Mass Spectrometry Analysis of the BCL-x_L_ Immunoprecipitates

**Mass spectrometry.** The BCL-x_L_ immunoprecipitated samples (i.e., BCL-x_L_-coated magnetic beads and bound proteins) were resolved on a 1D PAGE gel (Mini-PROTEAN^®^ TGXTM, Bio-Rad) at 200 V for 5 min and proteins were fixed on gels with Instant Blue stain (Abcam, Cambridge, UK). Following gel destaining with MilliQ water, the entire area of protein migration was excised and subjected to in-gel trypsin digestion, as previously described [21]. Peptides were extracted, dried in a Speed-Vac, subjected to desalting as previously described [21], and reconstituted in 12 μL of 2% acetonitrile (ACN), 0.1% formic acid for mass spectrometry analysis. Liquid chromatography with tandem mass spectrometry (LC-MS/MS) analysis was carried out as previously described with minor modifications [22]. For label-free proteomics analysis, the HPLC gradient was set to 98 min using a gradient that reached 30% ACN after 63 min, 34% after 66 min, and 79.2% after 71 min, following which there was an equilibration phase of 20 min at 2% ACN. Peptide sequences and protein identity were determined using MaxQuant software (version 1.6.0.1) by matching protein database of *P. falciparum* (UP000001450, release version 2016 04) and *Homo sapiens* (UP000005640, release version 2017 05) [21].

**Data analysis.** The outputs from MaxQuant were filtered to remove known contaminants, reverse sequences, and proteins identified by a single site. Protein identification FDR was as per default parameters of 0.01 [21], and label-free quantification was enabled. Match between run option was not enabled so that false positives would not be identified. Only proteins detected in at least 2 out of 3–5 biological replicates of uRBC- and iRBC-immunoprecipitated (IP) samples were considered for analysis. Proteins identified in the uRBC-IP or iRBC-IP samples and in the beads-only controls were excluded from the analysis. Interacting partners were based on the number of independent replicates the protein was identified in (2 or more) and the average intensities across all replicates for the specific proteins, which had to be greater than the treated (+inhibitor) samples (Tables 2 and 3). Of note, the relative intensity was used, not the normalized intensity [23], and the average did not include zero intensities in the case where no peptide was identified in a replicate.

## 3. Results

### 3.1. BCL-x_L_ Inhibitors Impair P. falciparum Proliferation

Six BCL-x_L_ inhibitors, selected based on their target specificity and/or current clinical use (Table 1), were tested in *in vitro* parasite proliferation inhibition assays. The selected BCL-x_L_ inhibitors include ABT-199 (Venclexta™, Venclyxto, Venetoclax), an FDA-approved compound for the treatment of leukaemia, and ABT-263 (Navitoclax), a compound currently in phase I/II clinical trials for the treatment of lymphoid malignancies. All six compounds significantly affected *P. falciparum* proliferation *in vitro*, as illustrated by the measured low IC_50_ values (Figure 1). ABT-263 and WEHI-539 displayed the lowest IC_50_ values of 0.74 and 0.82 µM, respectively, while A-1155463, ABT-199 and ABT-737 IC_50_ values were measured between 4 and 7 µM (4.23, 4.6 and 6.57 µM, respectively). A-1331852 presented the highest IC_50_ value at 14.05 µM.

### 3.2. BCL-x_L_ Inhibitors Induce Eryptosis of uRBCs and iRBCs

RBCs do not possess a classical apoptotic pathway due to the lack of a mitochondrion and a nucleus. Instead, they display a form of programmed cell death termed eryptosis [14]. Little is known about the molecular events leading to eryptosis, but a major hallmark of RBC death is the exposure of phosphatidylserine (PS) at the cell surface [33]. To test the ability of BCL-x_L_ inhibitors to induce eryptosis, we exposed uRBCs and *P. falciparum*-iRBCs to six BCL-x_L_ inhibitors (Table 1) for 4 h according to previously established guidelines [16], and RBC surface exposure of PS was measured by flow cytometry (Figure 1; treatment was with 5 µM or 10 µM of BCL-x_L_ inhibitor, depending on whether their IC_50_ was below or greater than 1 µM, respectively). A significant increase in the percentage of PS-exposing cells was observed in uRBCs in the presence of WEHI-539, A-1155463, and A-1331852 when compared with the respective no drug control. An increase in the percentage of PS-exposing cells was also observed in iRBCs in the presence of all six BCL-x_L_ inhibitors, albeit not significantly for ABT-737 (Figure 2). These results suggest that the inhibition of BCL-x_L_ induces eryptosis in uRBCs as well as in iRBCs.

### 3.3. Host BCL-x_L_ Is Recruited to the Parasite

In nucleated cells, the role of BCL-x_L_ in apoptosis is intrinsically dependent on its cellular localization, either at the mitochondria outer membrane or in the cytosol [34]. Therefore, the subcellular localization of BCL-x_L_ was investigated in iRBCs and uRBCs by Western blot and IFA (Figure 3).

Western blot analysis of saponin-lysed cells (see Figure 3A for a representative example) revealed the presence of BCL-x_L_ in the cytosol of uRBCs and iRBCs. However, BCL-x_L_ was also found in the membrane fraction of iRBCs, suggesting its association with the RBC plasma membrane, the parasitophorous vacuole membrane, the parasitophorous vacuole compartment, the parasite plasma membrane, and/or the parasite itself. The effectiveness of the cell lysis assay and control for protein leakage between the cytosolic and the membrane compartments was assessed with various antibody controls to detect RBCs cytosolic and membrane proteins, carbonic anhydrase I (CA-I), and protein 4.1, respectively. CA-I, a highly abundant cytosolic enzyme of RBCs [35], was detected in the cytosolic fractions of iRBCs and uRBCs but not in the membrane fractions, demonstrating that the membrane fractions were not contaminated with cytosolic proteins. Similarly, the major cytoskeletal protein of uRBCs, protein 4.1 [36], was detected in the membrane fractions of both uRBCs and iRBCs, but not in the cytosolic fractions, demonstrating that the cytosolic fractions were not contaminated with membrane or parasite proteins. Further, the absence of the *P. falciparum* protein PfHSP70.1 [37] in the uRBC fraction and in the RBC cytoplasm of infected cells indicated that parasite proteins do not contaminate the latter fraction. Of note, the signal of protein 4.1 was stronger in the membrane fraction compared to the total fraction in both uRBCs and iRBCs (although more striking in the iRBC sample). However, the signals of CA-I and PfHSP70.1 were consistent across the samples, confirming that equivalent amounts of proteins were loaded in each lane.

To gain further insight into the cellular localization of BCL-x_L_ in iRBCs, an IFA was performed using a human anti-BCL-x_L_ antibody. The BCL-x_L_ fluorescent signal produced was associated with the parasite, while BCL-x_L_ was not detected at the RBC membrane or in the cytosol of iRBCs (Figure 3B). In summary, we conclude that host BCL-x_L_ localizes to the uRBCs cytosol and is recruited to the vicinity of the parasite upon *P. falciparum* infection (see Discussion).

### 3.4. BCL-x_L_ Can Be Immunoprecipitated from uRBCs and iRBCs Cellular Extracts

In nucleated cells, the nature of the molecular complexes formed by BCL-x_L_ with different binding partners determines its pro- or anti-apoptotic activity. Typically, binding of BCL-x_L_ to the pore-forming proteins BAX and BAK leads to anti-apoptotic activity, whereas binding of BCL-x_L_ to the protein BAD leads to a pro-apoptotic outcome [38]. Interactors of BCL-x_L_ in RBCs have not been investigated to date. BCL-x_L_ from uRBCs and iRBCs was immunoprecipitated (IP) using a human anti-BCL-x_L_ antibody, and mass spectrometry was used to analyse the immunoprecipitated proteins (IP). To ensure the specificity of the BCL-x_L_ complexes, the BCL-x_L_ inhibitor WEHI-539 was used. Indeed, WEHI-539 has been shown to inhibit BCL-x_L_ in nucleated cells by displacing its binding partners from the BH3 domain-binding groove [27]. To test the hypothesis that WEHI-539 induces a similar displacement of BCL-x_L_ molecular complexes in uRBCs and iRBCs, cells were incubated with WEHI-539 (or with the control vehicle DMSO) prior to immunoprecipitation with an anti-BCL-x_L_ antibody coupled to Western blot and mass spectrometry analyses. (Appendix A). Cellular extracts incubated with beads alone (without antibody) were used as negative controls.

Western blot analysis of the BCL-x_L_ immunoprecipitates revealed the successful pull-down of BCL-x_L_ in the IP samples of uRBCs and iRBCs, in the presence and absence of the BCL-x_L_ inhibitor WEHI-539 (uRBC-IP +/− inhibitor, iRBC-IP +/− inhibitor; Appendix A). BCL-x_L_ was not detected in the negative controls (i.e., material recovered from beads alone, without BCL-x_L_ antibody) of uRBC and iRBC samples.

Mass spectrometry analysis of the BCL-x_L_ immunoprecipitates identified seven peptides (from 7 to 29 amino acids) from the human protein BCL2-L1 (Appendix A). Differential splicing of the *bcl2-l1* gene can produce either BCL-x_L_ or BCL-x_S_ proteins. BCL-x_S_ lacks amino acids 127 to 187 of BCL-x_L_; therefore, BCL-x_S_ lacks the BH1 domain and part of the BH2 domain [39]. This difference confers to BCL-x_S_ a different (and antagonist) role to that of BCL-x_L_. All peptides identified in this experiment shared 100% homology with BCL-x_L_ and importantly, two peptides (three and five) do not align with BCL-x_S_. Overall, 37% coverage of the BCL-x_L_ protein was achieved, including coverage of the BH1, BH3, and BH4 domains, which play a key role in the interaction with other members of the BCL-2 family [40] (Appendix A).

Calculation of the average intensity across all samples, as well as analysis of the number of BCL-x_L_ peptides identified per sample (Appendix A, respectively), revealed that a minimum of three peptides per replicate were detected in the IP samples, with intensities in the 10^7^ range. In contrast, either zero or one BCL-x_L_ peptides were identified in the negative controls, and when a peptide was detected, the corresponding intensity was low (~100× lower than in the immunoprecipitated samples). Overall, BCL-x_L_ was identified in immunoprecipitates prepared from uRBCs and iRBCs with a high level of confidence.

### 3.5. BCL-x_L_ Forms Molecular Complexes with μ-Calpain in uRBCs and with SHOC2 in iRBCs

Assuming that WEHI-539 displaces host BCL-x_L_ molecular complexes, the proteomics data set was analysed to identify proteins present in the BCL-x_L_ IP sample without an inhibitor (i.e., BCL-x_L_ binding partners), but absent or in low abundance in the BCL-x_L_ IP with an inhibitor (i.e., BCL-x_L_ displaced complexes). All proteins identified in the controls (i.e., extracts incubated with beads alone without BCL-x_L_ antibody) were excluded.

In uRBCs, μ-calpain (also called calpain-1), was identified in all three biological replicates from the uRBC-IP samples (i.e., uRBCs incubated with DMSO). However, μ-calpain was not detected in any of the four biological replicates from the uRBC-IP + inhibitor samples (i.e., uRBCs incubated with the BCL-x_L_ inhibitor WEHI-539) (Table 2). Therefore, this analysis identified μ-calpain as a specific BCL-x_L_ binding partner in uRBCs.

Conversely, in iRBCs, the human protein SHOC2, a leucine-rich repeat scaffold protein, was identified in three of the five independently generated iRBC-IP samples (i.e., iRBCs incubated with DMSO) but was not detected in any of the four BCL-x_L_ immunoprecipitates derived from iRBC-IP + inhibitor samples (i.e., iRBCs incubated with the BCL-x_L_ inhibitor WEHI-539) (Table 3). Therefore, human SHOC2 appears to form a molecular complex with host BCL-x_L_ in iRBCs. Further, the interaction between SHOC2 and BCL-x_L_ is specific to *P. falciparum* infection as SHOC2 was not immunoprecipitated in any of the uRBCs samples. Moreover, parasite proteins were also identified in the iRBC-IP samples, including the subunit D of a V-type proton ATPase, a pre-mRNA splicing factor, the pyrroline-5-carboxylate reductase, and the Apical sushi protein (Table 3). However, these proteins could only be detected in two or three of the five independent biological replicates, and with average intensities much lower than those obtained for the human proteins (i.e, the average intensity of the V-type proton ATPase is ~10× lower than that of the human SHOC2). Therefore, the *P. falciparum* proteins identified in the iRBC-IP samples do not represent convincing binding partner candidates of BCL-x_L_.

**Table 2 microorganisms-10-00824-t002:** **BCL-x_L_ binding partners identified in uRBCs.** Mass spectrometry analysis of BCL-x_L_ immunoprecipitates from uRBCs with and without the BCL-x_L_ inhibitor WEHI-539 over three and four independent biological replicates, respectively (*n* = 3–4). The number of replicates in which the protein was identified is indicated, along with its relative average intensity.

Protein	Gene Name	uRBC-IP (*n* = 3)	Average Intensity	uRBC-IP + Inhibitor (*n* = 4)	Average Intensity
Calpain-1 catalytic subunit	CAPN1	3	1,268,283	0	N/A
E3 ubiquitin-protein ligase ARIH2	ARIH2	2	787,200	0	N/A
Mannose-1-phosphate guanyltransferase	GMPPA	2	584,955	0	N/A
Biliverdin reductase A	BLVRA	2	576,585	1	355,100

## 4. Discussion

### 4.1. Host BCL-x_L_ Inhibition Impairs P. falciparum Growth

We have demonstrated that six BCL-x_L_ inhibitors impair *in vitro* proliferation of *P. falciparum*. In particular, the FDA-approved drug ABT-199 displayed an IC_50_ of 4.6 μM against parasite proliferation, which is within the range of the peak plasma concentrations found in patients treated for chronic lymphocytic leukaemia (CLL) or small lymphocytic lymphoma (SLL) [30]. Of note, anti-malarial drugs are generally found in the plasma at higher concentrations than their measured *in vitro* IC_50_ values, so *in vivo* studies will be required to investigate ABT-199 bioavailability before progressing toward clinical evaluation for malaria treatment.

Two other BCL-x_L_ inhibitors tested in this study, ABT-263, a drug currently in clinical trials for solid tumours [25,41], and WEHI-539, a highly specific inhibitor of BCL-x_L_, displayed the lowest *P. falciparum* IC_50_ values of 0.74 and 0.82 μM, respectively. This IC_50_ value of ABT-263 was around two- to eight-fold lower than the peak plasma concentrations measured following oral administration in an escalation study in humans [25]; however, future studies to measure plasma protein binding and free drug levels in plasma will be highly relevant to determine if this molecule can be further considered. Importantly, ABT-199 and ABT-263 half-lives are 19 h and 15 h, respectively. This feature is particularly attractive in the context of artemisinin-combination therapies where partner drugs (considering ABT-199 and ABT-263 in this case) are required to have longer half-lives than those of the artemisinins (i.e., greater than 10 h) [25,30,42]. This suggests that the pharmacokinetics of ABT-199 and ABT-263 may be sufficient to support further evaluation of the *in vivo* antimalarial efficacy of these candidates.

Interestingly, ABT-737 was previously found to impair the development of *P. yoelii* liver stages by inducing apoptosis of infected hepatocytes [11]. Similar to these previous findings, we demonstrated that BCL-x_L_ inhibitors induce PS exposure in iRBCs, more so than in uRBCs (Figure 2). These findings are in agreement with another previous study where inhibition of BCL-x_L_ with a BH3 peptide induced RBC death [13]. Of note, the three most specific inhibitors of BCL-x_L_ (WEHI-539, A-155463 and A-1331852) significantly induced eryptosis of uRBCs, while the other three did not (Figure 2). Taken together, our data indicate that BCL-x_L_ may play a role in eryptosis, although the molecular basis of this mechanism remains to be investigated.

*P. falciparum* has not been found to express BCL-x_L_ orthologs [43], suggesting that the BCL-x_L_ inhibitors tested in the present study act through the inhibition of host RBC factors, although off-target effects on other parasite proteins cannot be excluded at this stage. In human cells, some of the BCL-x_L_ inhibitors tested here also target other members of the BCL-2 family (BCL-2, BCL-x and BCL-w) in the nanomolar range (Table 1). Given that the *P. falciparum* inhibitory concentrations tested in this study were in the micromolar range, we cannot exclude the possibility that the antiparasitic activity observed is due to the inhibition of members of the BCL-2 family other than BCL-x_L_. However, BCL-2 is yet to be identified in RBCs, suggesting that the ABT compounds tested act through the specific inhibition of BCL-x_L_. In future, the use of reverse genetics to impair the expression of specific proteins in the erythroid lineage [44], prior to infection with *Plasmodium* spp., would assist in investigating off-target effects and formally assign a role for BCL-x_L_ in infection. Overall, we report that BCL-x_L_ inhibitors strongly impair the development of *P. falciparum* parasites *in vitro*.

### 4.2. BCL-x_L_ Is Recruited to the Parasite upon Infection

Considering that, in nucleated cells, the role of BCL-x_L_ is highly dependent on its cellular localization [34], we investigated the cellular localization of BCL-x_L_ in uRBCs and iRBCs and established that BCL-x_L_ is strictly cytosolic in uRBCs (Figure 3A). Our Western blot analysis revealed a stronger signal of the protein 4.1 in the membrane fraction of iRBCs compared to all other fractions (Figure 3A, middle panel). One possible explanation is that protein 4.1 associates with other human and/or parasite membrane proteins in iRBCs, although this would need to be further investigated. Regardless, CA-I and PfHSP70.1 panels demonstrated an equal load of proteins in each lane, therefore validating the detection of BCL-x_L_ in the membrane fraction of iRBCs. Of note, BCL-x_L_ has been previously reported as a membrane protein in RBCs [13]; however, in this study, the membrane fraction was not controlled for the presence of cytosolic proteins. Therefore, contamination of the membrane fraction by cytosolic BCL-x_L_ cannot be excluded.

Upon *P. falciparum* infection of RBCs, we have demonstrated that BCL-x_L_ continues to localize to the host RBC cytosol but that a pool of the protein relocates either to the parasitophorous vacuole membrane, the parasite membrane, or within the parasite itself (Figure 3). *P. falciparum* is known to export a considerable number of proteins to its host cell, including many targeted to the RBC membrane, but only a few examples of functional human proteins have been shown to be imported by the parasite [45,46,47,48]. Further work is required to identify the exact subcellular localization of BCL-x_L_ in iRBCs, and similar investigations should be carried out in infected hepatocytes. The BCL-x_L_ transmembrane domain (TM) has been shown to be a bona fide targeting signal for the mitochondria’s outer membrane [49]. Therefore, testing the competency of the BCL-x_L_ TM as a targeting signal to the parasitophorous vacuole membrane (or the parasite membrane) would consolidate our findings.

### 4.3. Identification of Novel BCL-x_L_ Binding Partners in RBCs

The classical binding partners of BCL-x_L_ (BAK, BAX, and BAD) have not been identified in RBCs in the current study following immunoprecipitation and mass spectrometry analyses. Instead, we found that BCL-x_L_ forms a molecular complex with the human protein μ-calpain in uRBCs and a complex with the human protein SHOC2 in iRBCs. Interestingly, the BCL-x_L_ molecular complexes formed in uRBCs (Table 2) are distinct to those formed in iRBCs (Table 3), highlighting the effect of *Plasmodium* infection on host factors and the existence of a host–parasite molecular crosstalk. Further, as the current investigations focused on the analysis of trophozoite stages, the nature of BCL-x_L_ molecular complexes should be investigated across the erythrocytic cycle (i.e., during ring and schizont stages).

Both the interactions of BCL-x_L_ with μ-calpain (in uRBCs) and with SHOC2 (in iRBCs) were disrupted in the presence of the BCL-x_L_ inhibitor WEHI-539, suggesting that both proteins bind to BCL-x_L_ via its BH3 domain_._ In addition, other BCL-x_L_ binding partners not displaced by the presence of the inhibitor WEHI-539 have also been identified (Appendix A). Presumably, such proteins bind to BCL-x_L_ outside of the BH3 domain. However, in this case, peptides were typically only identified in less than half of the replicates, at relatively low intensities, and many such proteins are not biologically relevant (e.g., human nuclear proteins). Therefore, the validity of such BCL-x_L_ binding partners warrants further investigation.

The most prominent BCL-x_L_ binding partner identified in RBCs in this study is μ-calpain (or calpain-1). μ-Calpain is a calcium-dependent cysteine protease involved in the cleavage of cytoskeletal proteins in apoptosis and eryptosis [50,51]. In platelets, μ-calpain also plays a role in the release of extracellular vesicles, exposing PS during platelet death induced by BCL-x_L_ inhibitors [52]. Of note, μ-calpain has been proposed to cleave the N-terminal end of BCL-x_L_ [53] (although this is disputed, and this cleavage is more often attributed to caspases [54,55]). The resulting cleaved-BCL-x_L_ is proposed to induce apoptosis. Western blot analysis conducted in this study did not detect cleaved BCL-x_L_ in either the uRBC or iRBC samples, despite the 16 kDa band being detected in the control sample (Figure 3), implying that μ-calpain binds to BCL-x_L_ in uRBCs without inducing protein cleavage. In addition to its role in cell death, μ-calpain has been shown to be hijacked by *P. falciparum* to assist with egress from the host RBC, with a re-localization from the RBC cytosol to the parasite membrane during the schizont stage [56], although BCL-x_L_ was not described to be involved.

The human protein SHOC2 was identified as a specific binding partner of BCL-x_L_ in iRBCs, as this study demonstrated that the BCL-x_L_-SHOC2 complex is disrupted by the BCL-x_L_ inhibitor WEHI-539 (Table 3). Noteworthy, SHOC2 was not found to bind to BCL-x_L_ in uRBCs (Table 2, Appendix A). These findings suggest that upon invasion, a BCL-x_L_–SHOC2 complex may perform a specific function relevant for parasite development.

SHOC2 is a scaffold protein containing many leucine-rich repeats that mediate protein–protein interactions. However, the role of SHOC2 in human RBCs remains unknown. In mice, *shoc2* knock-out leads to early embryonic lethality, and in zebra fish, its deletion results in important defects in erythropoiesis, suggesting a role of SHOC2 in RBC generation [57]. In nucleated cells, SHOC2 was found to be a positive regulator in the RAS/ERK/MEK kinase signalling cascade (or the Mitogen-Activated Protein Kinase, MAPK pathway) [58]. Interestingly, it has been previously shown that *P. falciparum* relies on host proteins within the MAPK pathway for its development, namely MEK [59] and Raf [60]. In the context of cancer treatment, it was found that a combination of BCL-x_L_ inhibitors together with MEK inhibitors could be efficient against KRAS mutant cancer models [61], highlighting cross-talk between the MAPK and the mitochondrion-dependent apoptosis pathways. However, the connection identified between these two pathways was the pro-apoptotic protein BIM, which to our knowledge has not yet been detected in RBCs. SHOC2 was also recently linked to DNA-damage-induced apoptosis, with *shoc2* knockdown affecting the BCL-2/BAX ratio (but no data on BCL-x_L_ were included in this study) [62]. Overall, our data are consistent with previous studies showing that the MAPK pathway is particularly important in *P. falciparum*-iRBCs and implicate BCL-x_L_ as a potential modulator of this pathway.

## 5. Conclusions

We have demonstrated that the inhibition of the host RBC BCL-x_L_ protein impairs *P. falciparum* growth *in vitro* and that the BCL-x_L_ inhibitors ABT-199 (an FDA-approved drug) and ABT-263 (a drug in clinical trials for solid tumours) display low micromolar and a submicromolar activity against the parasite, respectively. We have demonstrated that BCL-x_L_ is exclusively cytosolic in uRBCs but becomes recruited to the parasite/parasitophorous vacuole upon infection. Finally, we discovered that BCL-x_L_ forms novel molecular complexes in RBCs: μ-calpain was shown to interact with BCL-x_L_ in uRBCs, while SHOC2 binds to BCL-x_L_ in iRBCs. Although the functionality of BCL-x_L_ complexes in iRBCs remains to be fully elucidated, these complexes appear to play a key role in the development of *P. falciparum* inside human RBCs. Such findings present exciting possibilities for novel host pathways required for parasite development. Taken together, our findings suggest an important and novel role for human BCL-x_L_ in *P. falciparum* development. Although the role of human BCL-x_L_ both in uRBCs and in the development of *P. falciparum* remains unclear, we propose that BCL-x_L_ inhibition represents an interesting and novel antimalarial strategy. which would allow (i) host-targeted therapy opportunities, (ii) drug repurposing, and (iii) simultaneous targeting of parasite hepatic and blood stages.

## Figures and Tables

**Figure 1 microorganisms-10-00824-f001:**
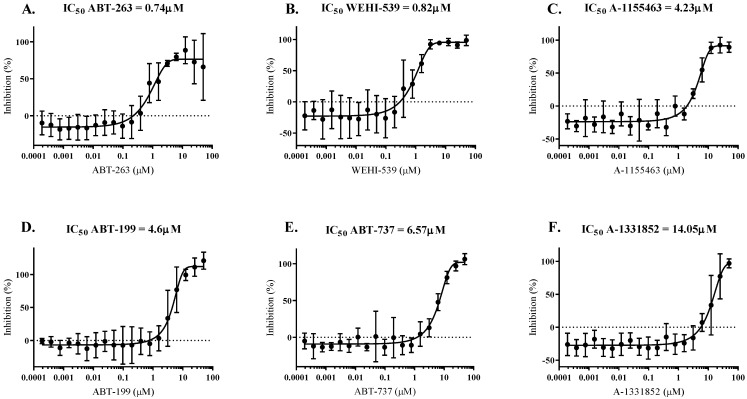
**Impact of BCL-x_L_ inhibitors on *P. falciparum* growth *in vitro*.** (**A**) ABT-263. (**B**) WEHI-539. (**C**) A-1155463. (**D**) ABT-199. (**E**) ABT-737. (**F**) A-1331852. Growth assays of six BCL-x_L_ inhibitors were conducted over 72 h on an asynchronous parasite culture, and the IC_50_ was calculated based on three independent biological replicates (*n* = 3, mean +/− SD).

**Figure 2 microorganisms-10-00824-f002:**
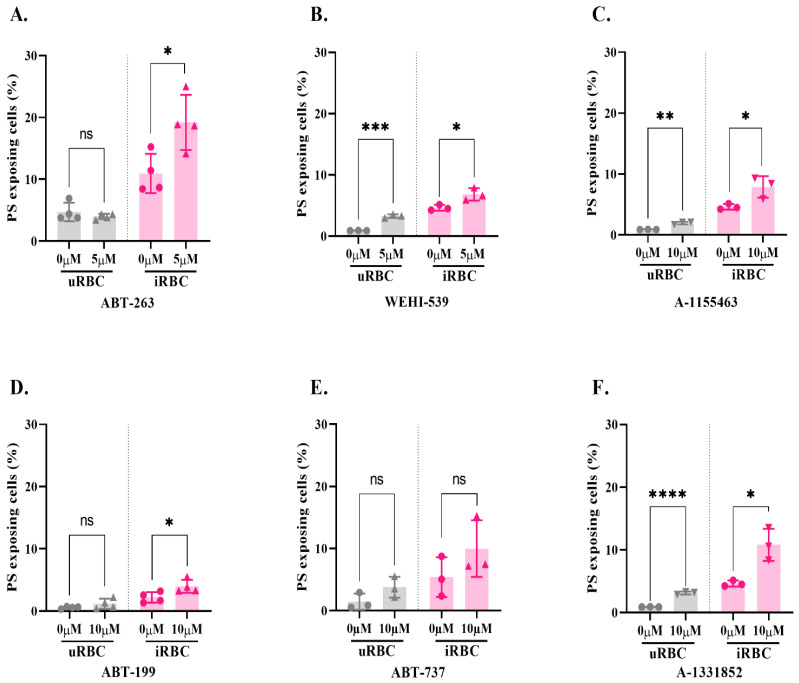
**PS exposure of uRBCs and iRBCs exposed to BCL-x_L_ inhibitors.** (**A**) ABT-263. (**B**) WEHI-539. (**C**) A-1155463. (**D**) ABT-199. (**E**) ABT-737. (**F**) A-1331852. iRBC or uRBC cultures were exposed to 5 or 10 µM of BCL-x_L_ inhibitors for 4 h in incomplete RPMI (following previously established guidelines [16]). The percentage of PS exposing cells was measured by flow cytometry (iRBCs were detected using a DNA stain). Individual values of *n* = 3–4 independent experiments (in technical duplicates) are shown, along with the mean +/- SD. Unpaired *t*-tests were conducted (****: *p* ≤ 0.0001; ***: *p* ≤ 0.001;**: *p* ≤ 0.01; *: *p* ≤ 0.05).

**Figure 3 microorganisms-10-00824-f003:**
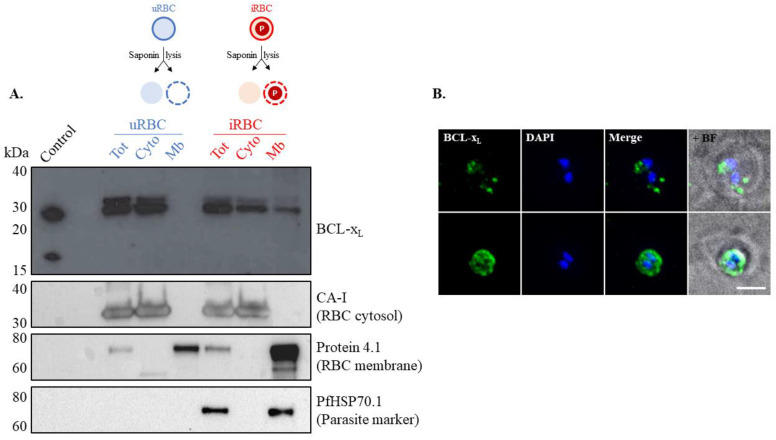
**Subcellular localization of BCL-x_L_ in uRBCs and iRBCs.** (**A**) Representative Western blot of uRBCs or iRBCs: total protein extraction (tot), cytosolic content (cyto—soluble fraction of a saponin lysis), and membrane fraction (Mb—insoluble fraction of a saponin lysis). BCL-x_L_ localizes exclusively to the cytosolic fraction in uRBCs and to the cytosolic and membrane fractions in iRBCs. A 16 kDa band corresponding to cleaved BCL-x_L_ is observed in the control but not in the uRBC and iRBC fractions. The control sample corresponds to protein extracts derived from the breast cancer cell line MDA-MB 231. The same blot was successively probed with the following antibodies: anti-BCL-x_L_, anti-Carbonic Anhydrase I (CA-I), anti-protein 4.1, and anti-PfHSP70.1. CA-I, Protein 4.1, and PfHSP70 are markers for the RBC cytosol, the RBC membrane, and the parasite, respectively. Note: control panels are also used in another study focussing on the protein BAD (to be published elsewhere) (**B**) Immunofluorescence assay of iRBCs (two representative cells are shown: early and mid-stage of infection in the upper and lower panels, respectively). BCL-x_L_ (green) is detected in the vicinity of the parasite, outside of the nucleus area stained with DAPI (blue) and inside the RBC membrane, which is visible by Bright Field (BF). The scale bar is 4 μm.

**Table 1 microorganisms-10-00824-t001:** **BCL-x_L_ inhibitors tested for inhibition of *P. falciparum* viability in this study.** The compound’s molecular target(s), current phase of clinical development, and clinical purpose are indicated. The reported activity of the compounds on *Plasmodium* liver stages is reported when available. The IC_50_ measured for each compound on *P. falciparum* blood stage in this study is reported (see Figure 1 for details). K_i_: inhibitory constant (reflective of the compound binding affinity).

	Human				*Plasmodium*	
Compound	Molecular Target	Clinical Development	Clinical Use	Peak Plasma Levels	Activity on Liver Stages	IC_50_ on Blood Stages (This Study)
**ABT-263** (Navitoclax)	BCL-2~BCL-x~BCL-w (K_i_ < 1 nM) [24]	Phase I/II [24]	lymphoid malignancies; Chronic lymphocytic leukaemia [24]	~5.75 μM [25]	No [26]	0.74 μM
**WEHI-539**	BCL-x_L_ (IC_50_ = 1.1 nM) [27]	Pre-clinical [27]	N/A (poor physicochemical properties) [27]	N/A	N/A	0.82 μM
**A-1155463**	BCL-x_L_ (K_i_ < 0.01 nM) [28]	Pre-clinical [28]	N/A	N/A	N/A	4.23 μM
**ABT-199** (Venclexta™, Venclyxto, Venetoclax)	BCL-2 (K_i_ < 0.01 nM) BCL-x_L_; BCL-w (K_i_ < 48 nM; <245 nM) [29]	FDA approved (2016)	Chronic lymphocytic leukaemia	~3.45 μM [30]	N/A	4.6 μM
**ABT-737**	BCL-2~BCL-x_L_~BCL-w (K_i_ < 1 nM) [31]	No	N/A (low solubility and bioavailability) [31]	N/A	Yes [11]	6.57 μM
**A-1331852**	BCL-x_L_ (K_i_ < 0.01 nM) BCL-w; BCL-2 (K_i_ = 4 nM; 6 nM) [32]	Pre-clinical [32]	N/A	N/A	N/A	14.05 μM

**Table 3 microorganisms-10-00824-t003:** **BCL-x_L_ binding partners identified in iRBCs.** Mass spectrometry analysis of BCL-x_L_ immunoprecipitates from iRBCs over a minimum of four independent biological replicates for each condition (iRBC-IP *n* = 4, iRBC-IP with inhibitor *n* = 5). The number of replicates in which the protein was identified is indicated, along with its relative average intensity.

**Human Proteins**					
**Protein**	**Gene Name**	**iRBC-IP** **(*n* = 5)**	**Average Intensity**	**iRBC-IP + Inhibitor** **(*n* = 4)**	**Average Intensity**
Leucine-rich repeat protein SHOC-2	SHOC2	3	21,690,633	0	N/A
Hydroxyacylglutathione hydrolase, mitochondrial	HAGH	2	3,275,720	0	N/A
T-complex protein 1 subunit zeta-2	CCT6B	2	1,400,900	0	N/A
Annexin	ANXA1	2	583,760	0	N/A
** *Plasmodium* ** **Proteins**					
**Protein**	**Gene ID**	**iRBC-IP** **(*n* = 5)**	**Average Intensity**	**iRBC-IP + Inhibitor** **(*n* = 4)**	**Average Intensity**
V-type proton ATPase subunit D	PF3D7_1341900	2	1,245,405	0	N/A
Pre-mRNA splicing factor	PF3D7_0922700	2	751,780	0	N/A
Pyrroline-5-carboxylate reductase	PF3D7_1357900	3	499,970	0	N/A
Apical sushi protein, ASP	PF3D7_0405900	2	371,175	0	N/A

## Data Availability

All the raw data and search result files (MaxQuant excel output) for the IP experiments have been deposited in the ProteomeXchange Consortium through the PRIDE partner repository [63] with identifier PXD032276. Username: reviewer_pxd032276@ebi.ac.uk, Password: MgxKbYIl.

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
