# Peer review of "Red Blood Cell BCL-xL Is Required for Plasmodium falciparum Survival: Insights into Host-Directed Malaria Therapies"

_microorganisms, 2022, doi:10.3390/microorganisms10040824_

Round 1
Reviewer 1 Report
The manuscript by Boulet et al studies the role of BCL-xL in RBCs and in P. falciparum infected RBCs by testing the effect of BCL-xL inhibitors on P. falciparum growth in RBCs. The authors conducted Western blotting, immunofluorescence experiments and mass spectrometry analysis of immunopurified BCL-xL protein complexes, and find that BCL-xL is relocated from RBC cytoplasm to the membrane fraction with the parasite. Distinct difference in host proteins are found for human μ-calpain that is specifically detected in RBCs and human SHOC2 is primarily identified in infected RBCs.
The experiments were well designed, and the study has been described well.
I have a few minor points for the authors to be addressed please.
1. Line 61 BCL -xl (B-cell lymphoma extra large)
2. Line 66. 6 BCL-xl inhibitors were selected for this study. Could the authors clarify why these 6 inhibitors have been selected and not others. Which specific properties of the selected inhibitors are different from other BCL-xl inhibitors.
3. Line 200: which type of label free proteomics was utilised in this study ? LFQ? Please provide a reference, and a definition for intensity that is used in the results section.
4. Line 203: FDR for protein identification is missing
5. Line 214: the average intensities across all replicates for the specific proteins, which had to be greater than the treated (+inhibitor) samples (Table 2 and 3).
Does this include any statistical test to identify significant changes in expression ?
Does this include zero intensities for replicates where proteins were not found ?
6. Material and Methods. A section on IC50 measurements seem to be missing
7. line 334: sequence coverage for BCL-xl protein is missing
8. line 337-334: I think a figure (heatmap or clustered heatmap) would support the text by visualising the expression differences for BCL-xl between IP samples.
9. Line 365-367: was the match between run option in MaxQuant enabled ? This will allow for more sensitive detection of low abundant peptides across all samples and may fill gabs that are currently present in the data set.
Author Response
We thank the reviewer for their positive feedback on our manuscript and constructive suggestions. We address below each of the reviewer’s questions/comments and where indicated have modified the manuscript accordingly.
- Line 61 BCL-xl (B-cell lymphoma extra large)
This sentence refers to the family of Bcl-2 proteins and not just to the Bcl-XL protein. Therefore, to clarify this, the sentence has now been modified and reads as follows (line 61): “ABT-737, an inhibitor of the anti-apoptotic members of the BCL-2 (B-cell lymphoma 2) family”
- Line 66. 6 BCL-xl inhibitors were selected for this study. Could the authors clarify why these 6 inhibitors have been selected and not others. Which specific properties of the selected inhibitors are different from other BCL-xl inhibitors.
To include the reviewer’s suggestion a new sentence has now been added to the Introduction and reads as follows (line 70-72): “The six BCL-xL inhibitors include one FDA-approved drug and have been selected based on their target specificity, clinical use and/or ongoing clinical trials.” Further justifications are provided in the Results section and in Table 1.
- Line 200: which type of label free proteomics was utilised in this study ? LFQ? Please provide a reference, and a definition for intensity that is used in the results section.
LFQ was enabled in MaxQuant, however the relative intensity value was used for the label free proteomics rather than normalised intensities. To address the reviewer’s suggestion, the Data Analysis subsection of the Material and Methods section has now been modified with additional information (line 221-223) and an added reference (https://doi.org/10.1021/acsami.1c19824; line 230).
- Line 203: FDR for protein identification is missing
Protein identification FDR was set to the default parameters of MaxQuant, which was 0.01. This has now been included in the text, see line 221-223.
- Line 214: the average intensities across all replicates for the specific proteins, which had to be greater than the treated (+inhibitor) samples (Table 2 and 3). Does this include any statistical test to identify significant changes in expression ? Does this include zero intensities for replicates where proteins were not found ?
No statistical test was conducted to compare intensities: the criteria used to select binding partners are explained line 220-226. The average intensities do not include zero values in the case where no peptide was identified, and a sentence has now been added at the end of the Data analysis section to clarify this (line 229-231): “Of note, the relative intensity, not normalized intensity, was used, and the average did not include zero intensities in the case where no peptide was identified in a replicate.”
- Material and Methods. A section on IC50 measurements seem to be missing
The methodology of the IC50 measurements is described in detail the Methods section entitled “Growth inhibition assays” (line 92). To highlight this, the title has now been modified accordingly and reads “Growth inhibition assays (IC50)”
- line 334: sequence coverage for BCL-xl protein is missing
We thank the reviewer for pointing this out. A sentence has now been added to the Results section (line 375): “37% coverage of the BCL-xL protein was achieved”. Further, we have added a new figure (Supplementary Figure S2B) to include the sequence of the BCL-xL protein and highlight the peptides that have been identified by mass spectrometry (as indicated in Supplementary Table S1 and Supplementary Figure S2A). Legend of Supplementary Figure S2B reads as follow (line 590-591): “Over a total of 233 amino acids, 86 amino acids were identified (peptide sequences highlighted in blue), corresponding to a 37% protein coverage.”
- line 337-334: I think a figure (heatmap or clustered heatmap) would support the text by visualising the expression differences for BCL-xl between IP samples.
We thank the reviewer for their suggestion of visual representation of the BCL-xL immunoprecipitation data. To clarify, the data presented in Supplementary Tables S2 and S3 and mentioned in the text lines 378-385 does not refer to differences in the expression of BCL-xL between samples. Instead, it reflects differences in the number of BCL-xL peptides (and their intensities) that have been successfully pulled down and identified by mass spectrometry, either in uRBCs (Supplementary Table S2), or in iRBCs (Supplementary Table S3). We therefore maintain that the tables (as currently presented in the supplementary information) best reflect the data, by providing all the information (i.e., number of peptides and protein average intensity).
- Line 365-367: was the match between run option in MaxQuant enabled ? This will allow for more sensitive detection of low abundant peptides across all samples and may fill gabs that are currently present in the data set.
The match between run option was not enabled in MaxQuant, as this is an IP experiment and we didn’t want any false positives due to incorrect fitting or noise. This has been clarified and a sentence has been added to the data analysis section (line 223): “Match between run option was not enabled, so that false positives would not be identified”.
Reviewer 2 Report
Authors are working with not much studied yet BCL-xL in red blood cells in uninfected and Plasmodium falciparum infected RBC, deepen in the BCL-xL role in the infection by Plasmodium falciparum. BCL inhibitors show the anti-parasitic effect, allowing to consider BCL-xL as target for antimalarial therapy. I want to underline the good quality of methodological description and correct Acknowledgements.
The minor critics are:
- Please, mention once the SHOC2 as Leucine Rich Repeat Scaffold Protein.
- Line 215: Table 2 and 3 are named before Table 1 (line 219)
- Figure 2A, the high basic level of apoptosis is shown for uRBC. It’s warrying: the approx. 4-5 % of apoptosis in all 4 samples and in the mean shown in Fig2A. This value theoretically must be the same as shown in the panels B-F with acceptable level of approx 1% of apoptotic RBC. Please, comment this. Then, could the high initial level of apoptosis be the reason for very high apoptosis level in ABT-263 treated iRBC?
- In this paper the conclusion of eryptosis in uRBC is based on three BCL-xl inhibitors, which are produced the significant effects on percentage of PS positive cells (Figure 2). But the other 3 inhibitors are not provoked apoptosis. Please, try to explain this in Discussion.
- Line 283, authors noted that protein 4.1 is present “more striking in the iRBC sample”, and in the Figure 3A this band is really very strong with small additional band at 50kDa. Supposing the equal quantity of proteins loaded in every line during electrophoresis, please try to explain the huge 4.1 protein quantity. Could it indicate the abundant protein quantity loaded in iRBC Mb line?
- Figure 3B and line 287: “The BCL-xl fluorescent signal was associated with parasite, …”. The figure 3B shown 2-3 DAPI positive nucleus in upper panel, and 3-4 DAPI positive nucleus in lover panel and massive green BCL-xl signal areas out of these parasites. Thus, the conclusion about “vicinity of the parasite” (line 290) is OK, instead “to the parasite itself” is overstating. Please, correct it.
- Table 2 vs Table 3: authors note the different human protein pattern of BCL-xl binding partners in uRBC vs iRBC. Please, speculate more in the Discussion why the pattern is completely different. Should we consider the trophozoite stage of iRBC in this experiment?
- Please, comment deeply the membrane localisation of BCL-xl, once described in [13] but was not seen in the present study.
- Line 462. The issue of cleaved BCL-xl at approx. 21kDa: to have the possibility to see it in the Figure 3 the range from 15-20 kDa to 37 kDa must be shown. In the actual presentation the reader can’t see the absence of truncated BCL-xl, thus it must be declared “not shown” instead of “Figure 3”.
Author Response
We thank the reviewer for their positive feedback on our manuscript and constructive suggestions. We address below each of the reviewer’s questions/comments and where indicated have modified the manuscript accordingly.
- Please, mention once the SHOC2 as Leucine Rich Repeat Scaffold Protein.
We thank the reviewer for the suggestion. SHOC2 is mentioned as a leucine rich repeat scaffold protein in the Introduction (line 78) and in Table 3 (page 11). For further clarification, this has now also been added to the Results section (line 397).
- Line 215: Table 2 and 3 are named before Table 1 (line 219)
Thank you for pointing this out. Indeed, a mention to Table 1 was missing in the Methods section and has now been added (line 94).
- Figure 2A, the high basic level of apoptosis is shown for uRBC. It’s warrying: the pprox.. 4-5 % of apoptosis in all 4 samples and in the mean shown in Fig2A. This value theoretically must be the same as shown in the panels B-F with acceptable level of pprox. 1% of apoptotic RBC. Please, comment this. Then, could the high initial level of apoptosis be the reason for very high apoptosis level in ABT-263 treated iRBC?
We agree with the reviewer that baseline eryptosis level in uRBCs in panel A of Figure 2 is higher than those in panels B-F. It is our experience (and others) that eryptosis levels can be variable depending on the blood donors and we have previously discussed this (Boulet et al., Front Cell Infect Microbiol 2021; doi:10.3389/fcimb.2021.630812). Therefore, eryptosis levels attributed to a given compound need to be directly compared with the no drug control within the same experiment. To emphasize this important argument the result section has now been modified accordingly and reads (line 280-282): “A significant increase in the percentage of PS-exposing cells was observed in uRBCs in the presence of WEHI-539, A-1155463 and A-1331852 when compared with the respective no drug control.”
- In this paper the conclusion of eryptosis in uRBC is based on three BCL-xl inhibitors, which are produced the significant effects on percentage of PS positive cells (Figure 2). But the other 3 inhibitors are not provoked apoptosis. Please, try to explain this in Discussion.
We agree with the reviewer that among the six BCL-xL inhibitors tested, three induced significant levels of eryptosis in uRBCs (WEHI-539, A-155463 and A-1331852; Figure 2). Based on previously published data, these three inhibitors display the lowest inhibitory constants (Ki) and the strongest specificity for BCL-xL. Therefore, to highlight this argument, a sentence has now been added to the Discussion (line 453-455) and reads: “Noteworthy, the three most specific inhibitors of BCL-xL (WEHI-539, A-155463 and A-1331852) significantly induced eryptosis of uRBCs, while the other three did not (Figure 2).”
- Line 283, authors noted that protein 4.1 is present “more striking in the iRBC sample”, and in the Figure 3A this band is really very strong with small additional band at 50kDa. Supposing the equal quantity of proteins loaded in every line during electrophoresis, please try to explain the huge 4.1 protein quantity. Could it indicate the abundant protein quantity loaded in iRBC Mb line?
Our western-blots have been carefully loaded with equal amounts of proteins in each lane and we argue the panels of CA-I and PfHsp70.1 demonstrate that (Figure 3A). However, we agree with the reviewer that the signal of protein 4.1 is surprisingly more intense in the iRBC membrane fraction than in the other fractions. To clarify this argument, a sentence has now been included in the Discussion section (line 474-480). We acknowledge that further investigations would be required to investigate the abundance of protein 4.1 in iRBCs, but that is outside of the scope of our manuscript, and it does not invalidate our findings regarding the presence of BCL-xL in the membrane fraction of iRBCs. We trust this statement will satisfy the reviewer.
- Figure 3B and line 287: “The BCL-xl fluorescent signal was associated with parasite, …”. The figure 3B shown 2-3 DAPI positive nucleus in upper panel, and 3-4 DAPI positive nucleus in lover panel and massive green BCL-xl signal areas out of these parasites. Thus, the conclusion about “vicinity of the parasite” (line 290) is OK, instead “to the parasite itself” is overstating. Please, correct it.
We thank the reviewer for this observation. The text has now been amended (line 324) and reads: “we conclude that host BCL-xL localizes to the uRBCs cytosol and is recruited to the vicinity of the parasite upon P. falciparum infection”
- Table 2 vs Table 3: authors note the different human protein pattern of BCL-xl binding partners in uRBC vs iRBC. Please, speculate more in the Discussion why the pattern is completely different. Should we consider the trophozoite stage of iRBC in this experiment?
We thank the reviewer for their suggestion. Accordingly, the following sentence has now been added to section 4.3 of the Discussion (line 500-506): “Interestingly, the BCL-xL molecular complexes formed in uRBCs (Table 2) are distinct to those formed in iRBCs (Table 3), highlighting the effect of Plasmodium infection on host factors and the existence of a host-parasite molecular crosstalk. Further, as the current investigations focused on the analysis of trophozoite stages, the nature of BCL-xL molecular complexes should be investigated across the erythrocytic cycle (i.e., during ring and schizont stages).”
- Please, comment deeply the membrane localisation of BCL-xl, once described in [13] but was not seen in the present study.
Section 4.2 has now been amended and reads (line 480-483): “BCL-xL has been previously reported as a membrane protein in RBCs [13]; however in this study the membrane fraction was not controlled for the presence of cytosolic proteins, therefore contamination of the membrane fraction by cytosolic BCL-xL cannot be excluded.”
- Line 462. The issue of cleaved BCL-xl at approx. 21kDa: to have the possibility to see it in the Figure 3 the range from 15-20 kDa to 37 kDa must be shown. In the actual presentation the reader can’t see the absence of truncated BCL-xl, thus it must be declared “not shown” instead of “Figure 3”.
We thank the reviewer for this recommendation. The upper panel of Figure 3A has now been amended to show a larger area of the BCL-xL blot (between 15 kDa and 40 kDa). This allows for the visualisation of cleaved of BCL-xL (16 kDa) in the control line, while the band is not detected in the uRBC and iRBC samples. Accordingly, the text in the Discussion section now reads (line 523-526): “Western blot analysis conducted in this study did not detect cleaved BCL-xL in either the uRBC or iRBC samples, despite the 16 kDa band being detected in the control sample (Figure 3)”. The legend of Figure 3A has also been amended accordingly (line 333).